# Learning Spatiotemporal Features via Video and Text Pair Discrimination

## Abstract

Current video representations heavily rely on learning from manually annotated video datasets which are time-consuming and expensive to acquire. We observe videos are naturally accompanied by abundant text information such as YouTube titles and Instagram captions. In this paper, we leverage this visual-textual connection to learn spatiotemporal features in an efficient weakly-supervised manner. We present a general cross-modal pair discrimination (CPD) framework to capture this correlation between a video and its associated text. We train our CPD models on both standard video dataset (Kinetics-210k) and uncurated web video dataset (Instagram-300k) to demonstrate its effectiveness. Without further fine-tuning, the learnt models obtain competitive results for action classification on Kinetics under the linear classification protocol. Moreover, our visual model provides an effective initialization to fine-tune on downstream tasks, which yields a remarkable performance gain for action recognition on UCF101 and HMDB51, compared with the existing state-of-the-art self-supervised training methods. In addition, our CPD demonstrates that pre-training a relatively small dataset is able to yield a comparable performance to those methods of using order magnitude more data, which is meaningful and practicable for the scenarios with limited computational facilities.

## 1 Introduction

Deep learning has made a remarkable progress for visual recognition in both image and video domain (Krizhevsky et al., 2012; He et al., 2016; Carreira & Zisserman, 2017; Feichtenhofer et al., 2018) by training powerful neural networks on large-scale manually annotated datasets (e.g., ImageNet (Deng et al., 2009) and Kinetics (Kay et al., 2017)). More importantly, it is well-established that this supervised pre-training on large-scale datasets would benefit the downstream tasks (e.g., object detection (Ren et al., 2015), pose estimation (He et al., 2017), and temporal action detection (Zhao et al., 2017)), in particular when the target datasets are relatively small. Yet, annotating a large-scale dataset for training such deep neural networks is costly and time-consuming, and even more challenging for video due to its various temporal structure and complex semantics. As a result, the existing video datasets size is still smaller than ImageNet in terms of training samples and classes. On the other hand, videos typically contain richer structure with abundant side information such as motion (Diba et al., 2019; Ng et al., 2018), audio (Arandjelovic & Zisserman, 2017; Korbar et al., 2018), and text (Miech et al., 2019; Sun et al., 2019b). So these expected these associated modalities are expected to provide useful cues to learn video representations in a more efficient way.

Language or text is probably the most natural and easy way to describe the semantic information of a video, and the associated textual information could be easily acquired when collecting video dataset (Rohrbach et al., 2017; Miech et al., 2019) from Internet or Movie. We argue that this correlation between a clip and its associated text could serve as an alternative supervision to learn **video representation** from scratch. This is different from some recent works (Sun et al., 2019b; Miech et al., 2019), in which these abundant textual information has been used to learn a **high-level visual-text embedding** applied to text-to-video retrieval or video captioning. Intuitively, it is more challenging to learn a general visual representation solely from text information without any human annotation, for reasons such as large numbers of noise in text, lacking careful initialization, and being hard to design an effective objective.

In this paper, we aim to learn effective video representation from noisy and diverse textual information, which could serves as the basis for a variety of downstream tasks. Basically, we learn a

mapping of text and video into a shared embedding space and leverage their correlation as supervision signal. The technical difficulty is how to design an effective objective function, that is capable of modeling this complex visual-textual correlation and as well easily optimized by training from scratch on noisy datasets. Inspired by unsupervised feature learning in images (Wu et al., 2018; Tian et al., 2019), we present a cross-modal pair discrimination (CPD) framework, which tries to recognize each video and text pair into a class via a non-parametric classifier. To solve the computational issues imposed by the huge numbers of pair classes, we adapt noise-contrastive estimation technique (Gutmann & Hyvärinen, 2010) to approximate the original loss function.

Specifically, we learn the CPD framework from web videos with the associated title or caption that could be directly crawled from web platforms such as YouTube (Kay et al., 2017) and Instagram (Duan et al., 2020). We utilize the off-the-shelf language models such as BERT (Devlin et al., 2019) or Word2vec (Mikolov et al., 2013) and devise a curriculum learning strategy to progressively train the video models. We first test the generalization ability of learned video representation by CPD on the Kinetics dataset (Kay et al., 2017) by using shallow classifiers such k-NN and linear classifier. It shows that our learned spatiotemporal features obtain promising results which are comparable to some supervised learning methods on the Kinetics dataset (Kay et al., 2017). Then, we investigate the generalization power of learned spatiotemporal features of CPD by fine-tuning on the Kinetics (Kay et al., 2017), UCF101 (Soomro et al., 2012) and HMDB51 (Kuehne et al., 2011) datasets, demonstrating that our method obtain superior performance to previous state-of-the-art self-supervised methods and comparable performance to the very recent methods of using orders of magnitude more videos (70M-100M vs. 0.3M).

## 2 RELATED WORK

**Self/Weakly Supervised Representation Learning.** Self supervised representation was popular in both image and video domains by designing various proxy tasks. In image domain, for instance, these tasks could be predicting the image context (Doersch et al., 2015), counting the objects (Noroozi et al., 2017), converting gray images to color one (Zhang et al., 2016), keeping global and local consistency (Hjelm et al., 2019). In video domain, typical examples include frame prediction (Diba et al., 2019; Vondrick et al., 2016), optical flow estimation (Ng et al., 2018; Zhou et al., 2017; Jayaraman & Grauman, 2017), instance tracking (Wang & Gupta, 2015; Wang et al., 2019b), temporal order or structure prediction (Misra et al., 2016; Fernando et al., 2017; Wei et al., 2018; Xu et al., 2019a). These learnt representations may capture some aspects of low-level image or video structures, but are generally outperformed by those using cross modal information.

Several cross-modal self-supervised tasks was proposed to enhance single-modality representation power and typical example is audio-visual representation learning (Aytar et al., 2016; Arandjelovic & Zisserman, 2017; Korbar et al., 2018). Meanwhile, some weakly-supervised methods were developed by utilizing web supervision obtained in an automatic way, such as query ID (Chen & Gupta, 2015; Ghadiyaram et al., 2019), and hashtag (Mahajan et al., 2018). Concurrent work (Miech et al., 2020) tried to learn video representations by using narration as supervision with instructional videos (e.g., HowTo100M (Miech et al., 2019)). However, they are limited by the video type. Our CPD is applicable to more general video type and we experiment with a much smaller dataset (0.3M vs. 100M) of both PGC and UGC videos, but achieves a similar performance on UCF101 and HMDB51. Concurrent work (Stroud et al., 2020) proposed a similar framework but required more training videos (0.3M vs. 70M) and richer textual information to obtain similar performance to ours.

**Motion, Audio, and Text.** Multi-modal information in videos provides natural cues for learning deep models. Motion or temporal information has been studied as to design proxy tasks to assist cross-modal learning, such as optical flow or tracking (Ng et al., 2018; Wang & Gupta, 2015), frame prediction (Diba et al., 2019; Vondrick et al., 2016), or high-level temporal structure (Wei et al., 2018; Xu et al., 2019a; Fernando et al., 2017). As most video contain synchronized audio and visual signals, audio information has served another common modality to supervised visual learning (Aytar et al., 2016; Arandjelovic & Zisserman, 2017; Korbar et al., 2018). However, both motion and audio information seem to be low-level signals and may lack high-level semantic for cross-modal learning.

Speech or text has been widely studied as another cross-modal setting in video learning (Sun et al., 2019b; Miech et al., 2019; Dong et al., 2019; Miech et al., 2018; Pan et al., 2016; Plummer et al., 2017). These works mainly aimed to learn a joint video-text embedding where visual and textual cues are adjacent if they are semantically. However, these works focused on learn high-level visual-

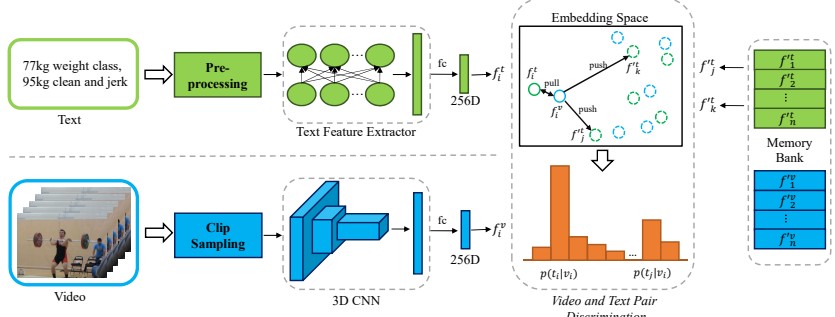

Figure 1: The pipeline of our cross-modal pair discrimination (CPD) framework. First, the visual and text are fed into modality-specific networks for feature extraction. Then, the visual and textual features are mapped into a common 256-dimensional space. The cross-modal framework is learned via video and text pair discrimination, which tries to make corresponding pairs closer than other inconsistent pairs using a softmax criteria. The learnt spatiotemporal features could be deployed directly or fine-tuned for downstream tasks.

textual embedding by using the off-the-shelf models as feature extractors. Instead, our proposed CPD framework addresses a different issue of video representation learning from scratch.

## 3 CROSS-MODAL PAIR DISCRIMINATION

In this section we provide an detailed description on our proposed *cross-modal pair discrimination* (CPD) for weakly supervised spatiotemporal feature learning. First, we present the whole framework and analyze its important properties. Then, we describe the training strategy of CPD framework. Finally, we introduce text and video feature extraction networks.

### 3.1 FRAMEWORK AND ANALYSIS

Our goal is to propose a weakly supervised representation learning method by exploiting the correlation between each video clip and its associated text information, which could be easily obtained from a variety of sources such as YouTube titles, Instagram captions and automatic speech recognition (ASR). It is generally assumed that these text information contains semantic information, but also might be noisy and irrelevant. Therefore, from technical perspective, we need to design an effective objective function and training strategy to capture this semantic correlation and as well also suppress the effect of noisy and irrelevant information. To this end, we devise a video-text pair discrimination objective and a curriculum learning strategy as follows.

More formally, as shown in Figure 1, we aim to learn a modality-specific embedding function $\mathcal{F}_v$ and $\mathcal{F}_t$ for the visual and textual information from a set of $N$ video clips and their associated textual information $\{(v_i, t_i)_{i=1}\}^N$. Let $\mathbf{f}_i^v$ and $\mathbf{f}_i^t$ denote $\mathcal{F}_v(v_i)$ and $\mathcal{F}_t(t_i)$, respectively. These embedding functions would map these two modality into a common space (i.e., $f_i^v \in \mathbb{R}^d$ and $f_i^v \in \mathbb{R}^d$), and related visual and text information should be close to each other. The embedding functions could be implemented by neural networks which will be clarified in next section. We first focus on how to devise objective function to optimize these embedding functions. Inspired by the work of unsupervised learning in images (Wu et al., 2018), we design a cross-modal pair discrimination objective to learn these two embedding functions.

**Self-instance discrimination.** In the original instance-level discrimination framework (Wu et al., 2018), each image is treated as a distinct class and it would learn a classifier to categorize each image into its own class. This framework could be naturally extended into the setting of video and text pair by directly using feature concatenation, and we call this extension as *self-instance discrimination*. Formally, this video-text level instance discrimination objective could be implemented with the following softmax criterion:

$$p(i|(v,t)) = \frac{\exp(\mathbf{w}_i^{vT}\mathbf{f}^v + \mathbf{w}_i^{tT}\mathbf{f}^t)}{\sum_{j=1}^{N}\exp(\mathbf{w}_j^{vT}\mathbf{f}^v + \mathbf{w}_j^{tT}\mathbf{f}^t)}, \tag{1}$$

where the $i^{th}$ video-text pair define a class $i$, $(\mathbf{w}_i^v, \mathbf{w}_i^t)$ is a weight for class $i$, and the class number is equal to training sample number $N$. This class weight represent a class prototype for each video-text instance and is probably not easy to optimize as we only have a single sample for each class. Thus, the above parametric classifier could be refined with the following non-parametric variant:

$$p(i|(v,t)) = \frac{\exp(\mathbf{f}_i^{vT}\mathbf{f}^v/\tau + \mathbf{f}_i^{tT}\mathbf{f}^t/\tau)}{\sum_{j=1}^{N}\exp(\mathbf{f}_j^{vT}\mathbf{f}^v/\tau + \mathbf{f}_j^{tT}\mathbf{f}^t/\tau)}, \tag{2}$$

where $\tau$ is a temperature parameter to control the class concentration level and our training objective is to optimize the likelihood $\prod_{i=1}^{N} p(i|(v_i, t_i))$. This straight forward extension shares the advantage of instance-level discrimination by directly modeling in the joint video-text space. Yet, in fact, the semantic information of text modality is higher than video pixels and we aims at learning video features with the supervision of textual information. To meet this requirement, we propose a refined objective function from the perspective of conditional distribution.

**Cross-pair discrimination.** According to the above analysis, we design the objective function by considering conditional distribution $p(i_t|v)$ and $p(i_v|t)$ rather than implicitly modeling distribution $p(v, t)$. Specifically, we design the following conditional distribution:

$$p(i_t|v) = \frac{\exp(\mathbf{f}_i^{tT}\mathbf{f}^v/\tau)}{\sum_{j=1}^{N}\exp(\mathbf{f}_j^{tT}\mathbf{f}^v/\tau)}, \tag{3}$$

where $i^{th}$ text define a text class $i_t$, and both $\mathbf{f}^t$ and $\mathbf{f}^v$ with unit-norm constraint. The conditional distribution $p(i_v|t)$ could be defined at the same way. We call this framework as *cross-pair discrimination*, and during training phase, the objective is to maximize the likelihood $\prod_{i=1}^{N} p(i_t|v_i) \prod_{i=1}^{N} p(i_v|t_i)$. The key difference between Equation (2) and (3) is that we propose to use cross-correlation term $\mathbf{f}^{tT}\mathbf{f}^v$ to replace the self-correlation term $(\mathbf{f}^{vT}\mathbf{f}^v + \mathbf{f}^{tT}\mathbf{f}^t)$. This cross correlation is more effective to capture the mutual information between visual and textual information, and thereby better at guiding the spatiotemporal feature learning from video with text information as supervision.

**Ranking loss.** There is some common ranking loss for cross-modal matching. To well study the effectiveness of proposed cross-modal pair discrimination objective, we also compare with a baseline of ranking loss, which is defined as follows:

$$\mathcal{L}(v_i, t_i) = \frac{1}{n-1}\sum_{j\neq i}\max(0, \delta + \mathcal{S}(\mathbf{f}_j^t, \mathbf{f}_i^v) - \mathcal{S}(\mathbf{f}_i^t, \mathbf{f}_i^v)), \tag{4}$$

where each video $v_i$ has a associated text $t_i$ and unrelated text $t_j$ from current batch. $\mathcal{S}(\mathbf{f}_j^t, \mathbf{f}_i^v)$ is the cosine similarity, $n$ is the batch size and $\delta$ is a margin. We apply Equation (4) in both ways of video with its associated text and text with its video. In experiment, we empirically compare this ranking loss with our designed cross-pair discrimination objective.

## 3.2 TRAINING CPD

The training of CPD framework needs to address two technical issues: (1) large number of video-text pair classes; (2) optimization difficulty on noisy video-text datasets by training from scratch.

**Noise-contrastive estimation.** In training stage, we adopt noise-contrastive estimation technique (Gutmann & Hyvärinen, 2010) to approximate Equation (3) to solve the computational issues by the huge numbers of pairs. The basic idea is to transform the multi-class classification problem in Equation (3) into a set of binary classification problem. In the binary classification task, the task is to distinguish between data sample and noise sample. The approximate training objective is to minimize the following loss function:

$$\mathcal{L} = -\mathbb{E}_{P(v)}\left\{\mathbb{E}_{P_d(i_t|v)}[\log h(i_t, v)] + m\mathbb{E}_{P_n(i_t'|v)}[\log(1 - h(i_t', v))]\right\}, \tag{5}$$

where $h(i_t, v) = \frac{p(i_t|v)}{p(i_t|v) + mp_n(i_t|v)}$, $P_d(i_t|v)$ is the actual data distribution and $P_n(i_t'|v)$ is the uniform distribution for noise, and $m$ denotes the noise frequency. To compute $p(i_t|v)$ efficiently and avoid large memory consumption, following (Wu et al., 2018), we maintain a **memory bank** to

store the visual and textual features for each training pair. The memory bank is updated dynamically during the training procedure.

**Curriculum learning.** To handle the optimization difficulty of directly training from scratch on noisy video-text dataset, we present a curriculum training strategy by resorting to the existing unsupervised pre-trained language models. To relieve the training difficulty, our curriculum learning strategy divides the training procedure into two stages. In the first stage, we fix the pre-trained language model and only update the parameters of visual model and embedding function. The motivation is that the language model is pre-trained well using corpus much larger than ours and the video model is totally trained from scratch. If we train both models simultaneously in the beginning, the random noise produced by video model will destroy the parameters of language model. In the second stage, after the good initialization of video model, we start to jointly train the visual-textual model with a smaller learning rate.

### 3.3 ARCHITECTURE DESIGN

**Video architecture.** For video representation, we use the 3D CNNs to extract spatiotemporal features from a video clip. Specifically, we randomly sample 8 frames from each video clip and sampling stride is 4. Following the implementation of slow stream in the recent SlowFast (Feichtenhofer et al., 2018), all filters from $conv_1$ to $res_3$ degenerate temporal convolutions into 2D convolution kernels and it only reserves 3D convolution kernels in $res_4$ and $res_5$ without temporal downsampling. We try two kinds of network architectures: (1) 3D ResNet34 trained on $112 \times 112 \times 8$ volumes and (2) 3D ResNet50 trained on $224 \times 224 \times 8$ volumes. The first tiny network is efficient for ablation study and then we transfer its optimal setting to the larger backbone and frame resolution. We also add a mapping layer to transform the visual features into 256-dimensional embedding space $\mathbf{f}^v$ and this 256-d vector is $\ell_2$-normalized.

**Text architecture.** Our textual stream subnetwork is based on the off-the-shelf language models. We choose Word2vec (Mikolov et al., 2013) and DistilBERT (Devlin et al., 2019; Sanh et al., 2019) as our textual encoders. Word2vec is an unsupervised word encoder, pre-trained by reconstructing the surrounding words of the continue sentences. We average word vectors which are 300 dimensional as textual encoder. BERT (Devlin et al., 2019) encodes long sentences by predicting the missing words given their bidirectional context, and DistilBERT achieves comparable performance with a faster and lighter model via knowledge distillation (Hinton et al., 2015). We average word embeddings of title generated by DistilBERT and obtain 768 dimensional text feature. Finally, two fully connected layers with ReLU and Batch Normalization (Ioffe & Szegedy, 2015) are added to our textual encoder to obtain textual feature $\mathbf{f}^t$ in the common embedding space, which is also $\ell_2$-normalized.

## 4 EXPERIMENTS

In this section, we present the experimental results of our proposed CPD framework. First, we describe the training and evaluation datasets with implementation details. Then, we conduct ablation study on our proposed CPD framework. Finally, we verify the effectiveness of CPD from two aspects: weakly-supervised representation learning and representation transfer.

### 4.1 DATASETS

In our experiment, we pre-train our CPD framework on two video-text datasets: Kinetics-210k (Kay et al., 2017) and Instagram-300k (Duan et al., 2020). Then, we fine-tune the video model on three human action datasets: Kinetics400 (Kay et al., 2017), UCF101 (Soomro et al., 2012) and HMDB51 (Kuehne et al., 2011).

**Kinetics-210k.** Following the recent self-supervised methods (Wang et al., 2019a; Korbar et al., 2018; Han et al., 2019), we utilize Kinetics (Kay et al., 2017) dataset for weakly-supervised pre-training of CPD. It is often called Kinetics400 since it has 400 action classes, but we count training video number as we do not use any class information for weakly-supervised representation learning. Due to invalid urls and data cleaning, the collected dataset contains around 210k video-text pairs, and thus we call this dataset as **Kinetics-210k**. To construct video-text pairs, we equip each clip with

the video title directly crawled from YouTube, termed as **Kinetics-title**. As the original title may be very noisy, we pre-process the text information in two ways. First, we delete special symbols and characters such as non-English words and emoji, termed as **Kinetics-title-clean**. Second, we use StanfordNLP (Qi et al., 2018) to obtain the dependency tree of sentences in titles and only reserve verbs and nouns, named **Kinetics-title-tree**.

**Instagram-300k.** To avoid data bias in Kinetics caused by human annotation (i.e., trimmed videos with an action), we further verify the effectiveness our CPD model on an uncurated web video dataset (Duan et al., 2020). This new dataset is constructed from Instagram by searching action label of Kinetics-400 but without any manual filtering. *Due to limited computation resource and also for fair comparison with pretraining on Kinetics-210k*, we randomly sample 300k from the original web video dataset, termed as **Instagram-300k**.

An important difference is that the these videos are with User Generated Content (UGC) and accompanied by captions uploaded by users. Therefore, its video content distribution is much different with those in Profession Generated Content (PGC) in UCF101 and HMDB51, and the text noise is also much higher. So, it is more challenging to train a pre-trained CPD model on Instagram-300k.

**UCF101 and HMDB51.** We evaluate the generalization of our pre-trained models by fine-tuning on two small human action datasets: UCF101 (Soomro et al., 2012) and HMDB51 (Kuehne et al., 2011), which contain 13k videos of 101 classes and 7k video of 51 classes respectively. We report ablation study on the first split and report average performance over three splits for fair comparison.

## 4.2 Implementation details

**Weakly supervised learning of CPD.** We train our CPD model on video-text datasets and use video-text retrieval on 1k unseen video-text pairs as validation set duration training. Specifically, 8 frames are sampled from each video clip and the sampling stride is 4. We use SGD to optimize our objective and the training parameters include a momentum of 0.9 and 1e-4 for weight decay. We set temperature parameter $\tau = 0.07$ and noise frequency $m$ to 4096. In the beginning, we fix the pre-trained language model and the learning rate is set as 0.2. When the retrieval performance on validation set saturates (170 epochs for 3D ResNet34 and 110 epochs for 3D ResNet50), we start to update the language model with learning rate of 3e-5 and decrease the rest learning rate to 0.02. The maximize training number is 250 epochs. For input size of $112 \times 112 \times 8$ , the mini-batch size is 64 clips per GPUs and 16 clips per GPUs for input size of $224 \times 224 \times 8$. We use 8 GPUs for training.

**Evaluation on representation learning.** We first verify our CPD learned representation by employing a shallow classifier on *frozen features*. Specifically, we utilize k-Nearest Neighbor (kNN) and linear classifier based on extracted features for classification. For video feature extraction, we sample 10 clips from each video and each clip contains 8 frames with 4 sampling stride. The 256-dimensional embedding feature and the output of global average pooling are extracted as features. The extracted features over 10 clips in a video are averaged as a video-level representation. We choose cosine distance as distance metric in kNN and set $k = 25$. As for linear classifier, a fully connected layer after Batch Normalization is added with cross-entropy loss. We adopt Adam with learning rate of 1e-3 and reduce by a factor of 10 every 10 epochs, stopping at 30 epochs.

**Evaluation on representation transfer.** A main goal of representation learning is to transfer them to downstream tasks. We *fine-tune* the learned spatiotemporal representation on the UCF101, HMDB51 and a small fraction of Kinetics400. During fine-tuning, 16 frames with stride 4 are sampled as input. We simply replace the embedding layer of video model with a new fully-connected layer and multi-way softmax for action recognition. The classifier is trained using the SGD optimizer with an initial learning rate 1e-2 and weight decay 5e-4. Learning rate is decreased twice by a factor of 10 when the validation loss saturates. During testing, for each video, we uniformly sample 10 clips and each clip contains 3 crops, following the common practice (Feichtenhofer et al., 2018).

## 4.3 Ablation study

In this study, we pre-train our CPD models on Kinetics-210k dataset and choose the task of representation transfer by fine tuning on UCF101 split 1 for evaluation.

| Objective function | Accuracy(%) |
|---|---|
| Random init. | 50.0 |
| Ranking loss | 79.9 |
| Self-instance Dis. | 51.1 |
| Cross-pair Dis. | **82.2** |

(a) Study on loss functions.

| Training strategy | Accuracy(%) |
|---|---|
| Random init. | 50.0 |
| Direct fine-tuning | 81.3 |
| Curr. learning1 | 82.2 |
| Curr. learning2 | **84.2** |

(b) Study on training strategies.

| Textual encoder | Data | Accuracy(%) |
|---|---|---|
| Random Init. | - | 50.0 |
| Word2vec | Tree | 83.1 |
| DistilBERT | Tree | 82.1 |
| Word2vec | Clean | 82.5 |
| DistilBERT | Clean | **84.2** |

(c) Study on textual encoders.

Table 1: **Ablation study** on UCF101 by fine tuning a pre-trained CPD model from Kinetics-210k.

| Backbone | Pre-trained Sup. | Layer (Dim) | kNN | LC |
|---|---|---|---|---|
| 3D-ConvNet (Kay et al., 2017) | Kinetics-400 Label | - | - | 56.1 |
| 3D ResNet34 (Hara et al., 2018) | Kinetics-400 Label | - | - | 60.1 |
| 3D ResNet50 (ours) | Kinetics-400 Label | - | - | 73.2 |
| ResNet50 | ImageNet Label | res5 (2048) | 42.8 | 56.1 |
| 3D ResNet34 | Instagram-300k Caption | emb (256) | 34.5 | 37.3 |
| 3D ResNet34 | Instagram-300k Caption | res5 (512) | 36.1 | 44.6 |
| 3D ResNet50 | Instagram-300k Caption | emb (256) | 51.1 | 51.7 |
| 3D ResNet50 | Instagram-300k Caption | res5 (2048) | 51.1 | 55.4 |
| 3D ResNet34 | Kinetics-210k Title | emb (256) | 49.9 | 50.8 |
| 3D ResNet34 | Kinetics-210k Title | res5 (512) | 50.1 | 53.3 |
| 3D ResNet50 | Kinetics-210k Title | emb (256) | 58.0 | 59.6 |
| 3D ResNet50 | Kinetics-210k Title | res5 (2048) | **58.2** | **63.8** |

Table 2: **Evaluation on weakly-supervised representation learning without fine-tuning.** Top-1 classification accuracy is reported on Kinetics-400 validation set.

**Objective function.** We compare three objective functions for cross-modal pair discrimination described in Section 3.1. We pre-train models by utilizing DistilBERT as textual encoder without fine-tuning and the experimental results are reported in Table 1a. Self-instance discrimination almost has no contribution to learn effective representation as there is no cross-modal correlation modeling. Cross-pair discrimination gives a better performance than ranking loss as cross-pair discrimination can construct negative video-text pairs from entire dataset while ranking loss is only optimized by negative pairs from current batch. More theoretical analysis can be found in Section. A.1 of the Appendix.

**Curriculum learning.** We design different training strategies from noisy video-text datasets. The first strategy is to fine-tune the pre-trained textual encoder directly at the beginning. Then we compare with stage I and stage II of curriculum learning proposed in Section 3.2. All these strategies are pre-trained on Kinetics-title-clean. The numerical results are summarized in Table 1b. Fixing the pre-trained language model gives better performance than direct fine-tuning at the beginning (+0.9%). We ascribe this to the fact that the random noise produced by video model destroy the well pre-trained textual encoder at the beginning. Also, fine-tuning the language model after the video model is well initialized further boost the accuracy by 2.0%.

**Different textual information.** In this experiment, we choose video-text pairs from Kinetics-title-tree, Kinetics-title-clean datasets and utilize Word2vec and DistilBERT as a textual extractor. The experimental results are reported in Table 1c. For textual encoder, abundant and video-specific text information benefits to train our CPD model with stronger language model such as DistilBERT according to the performance difference between Kinetics-title-tree and Kinetics-title-clean (82.1% vs. 84.2%). As for shallow textual encoder (e.g., Word2vec), simple text information from Kinetics-title-tree dataset gives better performance than abundant text information (83.1% vs. 82.5%). From above observation, it can be concluded that Word2vec is more good at concise and accurate text while DistilBERT can handle more complex and noisy sentences which is close to realistic setting. Also, it is affordable to utilize strong language models due to our curriculum learning strategy and lightweight DistilBERT model.

## 4.4 EVALUATION ON REPRESENTATION LEARNING

To evaluate our learned representation, we report the classification performance on validation set of Kinetics via training shallow classifiers on *frozen features* as shown in Table 4.3. We perform kNN classifiers and linear classifiers (LC) on the embedding features or visual features from global average pooling after *res5*. In this shallow learning setting, we also compare with ImageNet pre-training representation (ResNet50) by using the same classifier. *First*, the representation learnt from

| Method | Supervision | Backbone | Pre-trained Dataset | UCF101 | HMDB51 |
|---|---|---|---|---|---|
| Random Init. (Hara et al., 2018) | - | 3D ResNet18 | - | 42.4 | 17.1 |
| Kinetics Pre-trained (Hara et al., 2018) | Action label | 3D ResNet50 | Kinetics | 89.3 | 61.0 |
| Supervised SOTA (Xie et al., 2018) | Action label | S3D | Kinetics | 96.8 | 75.9 |
| Shuffle & Learn (Misra et al., 2016) | Order verification | CaffeNet | UCF101/HMDB51 | 50.2 | 18.1 |
| OPN (Lee et al., 2017) | Sequence order | VGGNet | UCF101/HMDB51 | 59.8 | 23.8 |
| CMC (Tian et al., 2019) | Optical flow | CaffeNet | UCF101 | 55.3 | - |
| O3N (Fernando et al., 2017) | Odd-one-out | AlexNet | UCF101 | 60.3 | 32.5 |
| MASN (Wang et al., 2019a) | Motion | C3D | Kinetics-400 | 61.2 | 33.4 |
| COP (Xu et al., 2019b) | Clip order | 3D ResNet10 | UCF101 | 64.9 | 29.5 |
| DPC (Han et al., 2019) | Prediction | 3D ResNet34 | Kinetics-400 | 75.7 | 35.7 |
| CBT (Sun et al., 2019a) | Audio(Text)/Context | S3D | Kinetics-600 | 79.5 | 44.6 |
| AVTS (Korbar et al., 2018) | Audio | I3D | Kinetics-600 | 83.7 | 53.0 |
| AVTS (Korbar et al., 2018) | Audio | MC3 | Audioset-1.8M | 89.0 | 61.6 |
| XDC (Alwassel et al., 2019) | Audio | R(2+1)D | Kinetics-400 | 84.2 | 47.1 |
| XDC (Alwassel et al., 2019) | Audio | R(2+1)D | IG-65M | **91.5** | 63.1 |
| MIL-NCE (Miech et al., 2020) | Audio(Text) | S3D | HT-100M | 91.3 | 61.0 |
| TWS (Stroud et al., 2020) | Text (Title, Des, Tag etc.) | S3D-G | WVT-70M | 90.3 | **65.3** |
| CPD (Ours) | Caption | 3D ResNet50 | Instagram300k | 89.9 | 63.8 |
| CPD (Ours) | Title | 3D ResNet50 | Kinetics210k | **90.5** | 63.6 |

Table 3: **Evaluation on representation transfer by fine-tuning**. We compare our CPD model with other methods trained on different type of supervision.

Kinetics-210k generally outperforms that of Instagram-300k. The reason could be ascribed to the video distribution gap between UGC (Instagram) and PGC (Youtube), and also much noisier textual information in Instagram-300k. *Second*, we compare with ImageNet pretrained features, and our CPD representation is better under the same backbone. *Finally*, we compare with some end-to-end trained representations with action labels, and there is still a performance gap between our representation and supervised end-to-end representation (e.g. 63.8% vs. 73.2%).

### 4.5 EVALUATION ON REPRESENTATION TRANSFER

Transferring learned representation to downstream tasks is a main goal of representation learning. We transfer them to action recognition task on small datasets, namely UCF101 and HMDB51. We compare our CPD model pre-trained on Instagram-300k and Kinetics-210k with a randomly initialized network, self-supervised methods solely based on visual information, including Shuffle & Learn (Misra et al., 2016), CMC (Tian et al., 2019), MASN (Wang et al., 2019a), COP (Xu et al., 2019b), DPC (Han et al., 2019) and so on, and representation learning methods based on multi-modal information (e.g., audio, text), including CBT (Sun et al., 2019a), AVTS (Korbar et al., 2018), XDC (Alwassel et al., 2019), MIL-NCE (Stroud et al., 2020), and TWS (Stroud et al., 2020).

As shown in Table 3, our CPD models generally outperform those self-supervised learning approaches of only using visual information ($\geq 10\%$ on UCF101 and $\geq 20\%$ on HMDB51), which indicates that cross-modal information is useful cue for visual representation learning. Meanwhile, our CPD representations obtain comparable performance to the concurrent works (i.e., MIL-NCE and TWS) of using text as weak supervision. However, our CPD uses a much smaller pre-training dataset of around 0.3M videos, while the other methods uses 70M-100M videos. Training a CPD model on a such large-scale dataset is almost impossible for a university lab with limited computational facilities. *Our work demonstrates that pre-training a relatively small video-text dataset is also possible to match the SOTA performance, and this is quite meaningful and practicable for university lab.* Finally, we notice that the gap of CPD models learned from Instagram-300k and Kinetics-200k is very small, indicating that our CPD model can effectively handle high noise in text.

## 5 CONCLUSION

In this paper, we have presented a general cross-modal pair discrimination (CPD) framework to capture the correlation between a video clip and its associated text from real word and adopt noise-contrastive estimation to approximate the objective. Without fine-tuning, the learned models obtain competitive results for action classification on Kinetics dataset with a shallow classifier. Also, our visual models provide an effective initialization to fine-tune on the datasets of downstream task, and matches the state-of-the-art performance with a much smaller pre-training dataset.

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

## A    TRAINING DETAILS OF CPD

we adopt noise-contrastive estimation technique (NCE) to approximate objective function in Equation (5) in our main paper. The purpose of NCE is to transform the multi-class classification problem into a set of binary classification problems by comparing data distribution against noise distribution. So $p_n$ is noise distribution and we formalize it as a uniform distribution: $p_n = \frac{1}{N}$, where $N$ is the number of video-text pairs. $h(i_t, v)$ is the posterior probability of feature from the data distribution which means video and text are matched. $m$ is the number of negative pairs and we set it as 4096. For each video feature $\mathbf{f}^v$, we take its related text feature $\mathbf{f}^t_i$ and sample 4096 unrelated text features $\mathbf{f}^t_j$ which are all from memory bank. The FPS of training videos are 30. The code of CPD will be released.

### A.1    ANALYSIS ON DIFFERENT LOSS FUNCTIONS

More insight about why our loss is better than ranking loss could be found from gradient back-propagation. Let $\mathbf{f}^{t+}$ and $\mathbf{f}^{t-}$ represent the associated and unrelated text feature. For ranking loss, the negative gradient w.r.t $\mathbf{f}^v$ is $\mathbf{f}^{t+} - \mathbf{f}^{t-}$ if $\mathcal{L} > 0$ else 0. For CPD loss, it is $[1 - h(i_t^+, v)]/\tau \mathbf{f}^{t+} - \sum h(i_t^-, v)/\tau \mathbf{f}^{t-}$. We observe our loss assign different weights to different examples based on their posterior probability $h$, which helps learn from hard examples while the ranking loss treats them equally.

## B    REPRESENTATION TRANSFER ON KINETICS

| Method | The Amount of Labeled Data | | |
|---|---|---|---|
| | 1% | 10% | 20% |
| From scratch | 0.3 | 10.7 | 33.3 |
| ImageNet Inflation | 12.8 | 36.8 | 45.7 |
| Ours (Instagram-300k) | 18.7 | 41.3 | 47.4 |
| Ours (Kinetics-210k) | 25.9 | 43.1 | 47.8 |

Table 4: Results of classification with small amount of labeled data on Kinetics-400 validation set (showing top-1 accuracy). We utilize 3D ResNet34 as backbone and pre-train it on Kinetics-210k and Instagram-210k.

Our weakly-supervised pre-trained representation can be an efficient initialization when training the model with only a small amount of labeled data. We randomly choose a small fraction of Kinetics-400 training set as labeled data and fine-tune the pre-trained model on it. We report the performance of top-1 accuracy which is trained on labeled subset of 1%, 10% and 20% of the entire dataset in Table 4. We compare our method with training from scratch and ImageNet inflated model as baselines. Our method significantly surpasses the baselines on all present proportion of labeled subset especially when the amount of labeled data is extremely small. When only 1% of data is labeled, training from scratch can not learn anything yet our model achieves 18.7% and 25.9% top-1 accuracy. Both our CPD pre-trained models on Instagram and Kinetics outperform the ImageNet pre-trained models.

## C    EVALUATION ON ZERO-SHOT CLASSIFICATION

We evaluate our visual-textual embedding of CPD model with zero-shot classification on UCF101 and Kinetics-400 without any fine-tuning in Table 5. We transform class labels and video clips into the same embedding space and recognize the video clip to its closest class with cosine distance. We compare our method with Mettes *et al.* (Mettes & Snoek, 2017) which realizes zero-shot localization and classification of human action in video via spatial-aware object embeddings on UCF101. Following (Mettes & Snoek, 2017), we select different classes for 10 times and average their accuracies for testing except the class number is 101. We outperform for every number of testing classes. For Kinetics-400, we achieve top-1 accuracy of 43.7% without fine-tuning and training label. In addition, top-1 accuracy of 20 random classes reaches to 74.4%, which shows the strong capability of our visual-textual embedding.

| Methods | UCF-101 | | | | Kinetics-400 | | | |
|---|---|---|---|---|---|---|---|---|
| | Train | Test | Split | Acc. | Train | Test | Split | Acc. |
| Mettes (Mettes & Snoek, 2017) | - | 101 | 3 | 32.8 | - | - | - | - |
| Ours(3D ResNet34) | - | 101 | 3 | 40.6 | - | 400 | 1 | 38.2 |
| Ours(3D ResNet50) | - | 101 | 3 | 39.9 | - | 400 | 1 | 43.7 |
| Mettes (Mettes & Snoek, 2017) | - | 50 | 10 | 40.4 | - | - | - | - |
| Ours(3D ResNet34) | - | 50 | 10 | 47.2 | - | 100 | 10 | 55.3 |
| Ours(3D ResNet50) | - | 50 | 10 | 44.8 | - | 100 | 10 | 57.4 |
| Mettes (Mettes & Snoek, 2017) | - | 20 | 10 | 51.2 | - | - | - | - |
| Ours(3D ResNet34) | - | 20 | 10 | 54.4 | - | 20 | 10 | 73.1 |
| Ours(3D ResNet50) | - | 20 | 10 | 58.1 | - | 20 | 10 | 74.4 |

Table 5: Top-1 accuracy of zero-shot classification on UCF-101 and Kinetics-400. We outperform other methods without any extra labeled data and training procedure after pre-training on Kinetics-210k.

## D ANALYZE TEXT INFORMATION

### D.1 ANALYSIS ON KINETICS TITLE

| Datasets | At Least One(%) | All(%) | Rel(%) |
|---|---|---|---|
| Kinetics-title-tree | 90.5 | 44.3 | 46.3 |
| Kinetics-title-clean | 91.6 | 38.4 | 26.0 |

Table 6: Analyze text information of Kinetics-210k datasets. *At Least One*: The proportion of text information that contains at least one word in action classes of Kinetics-400. *All*: The proportion of text information that contains the entire action class. *Rel*: The proportion of word in text information that is relevant to action classes.

We provide an analysis of text information we used and the result in Table 6. First, there exists a large overlap between action class and text information (more than 90% for at least one word and more than 38% for complete action class). However, the titles also contain many other words and noisier information than action classes. Only 26% of words in Kinetics-title-clean are relevant to action classes.

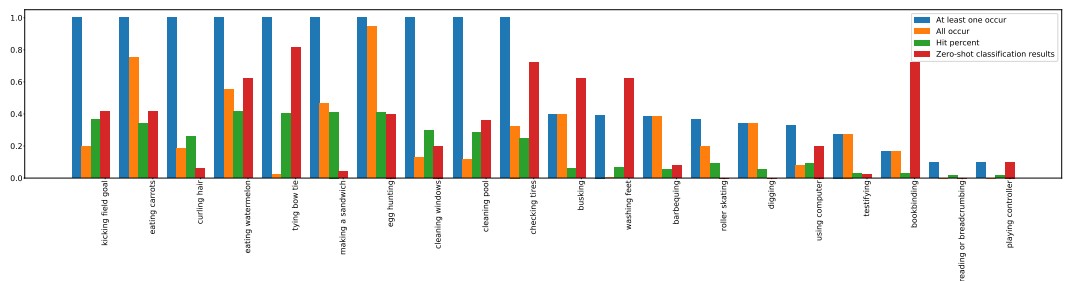

Figure 2: List of top 10 and bottom 10 kinetics classes sorted by the frequency of at least one word in label occurring in according title of Kinetics-title-clean dataset. Zoom in for more details.
We also report the per-class accuracy of top 10 and bottom 10 classes sorted by word overlapping in Figure 2 and see that this accuracy is not positively correlated with word overlapping percentage. Finally, we provide some examples of videos and their titles from Kinetics-210k in Figure 3.

### D.2 VISUALIZATION OF INSTAGRAM CAPTION

Since videos from Instagram-300k are not annotated or filtered by human, both of their visual and textual information are very noisy. Figure 4 demonstrates some examples of videos and their associated captions. Figure 4a presents an example of high-quality video and relative accurate caption that both are about folding napkin. Many captions describe some useful information but also contain noisy text that is not related to video content (e.g., *summerdays and gettingtattooed* in Figure 4b and very long sentences in Figure 4d). In addition, there are some correct but not totally accurate

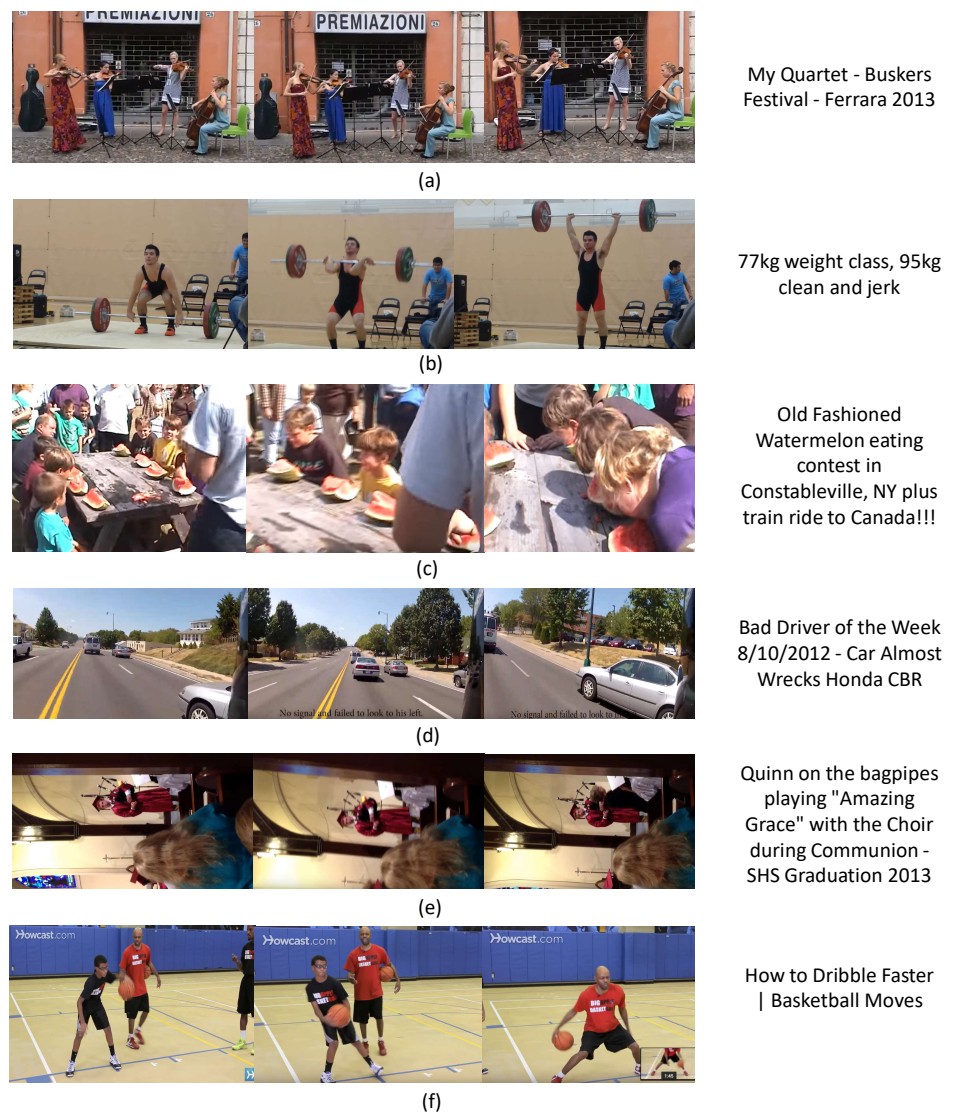

My Quartet - Buskers Festival - Ferrara 2013

(a)

77kg weight class, 95kg clean and jerk

(b)

Old Fashioned Watermelon eating contest in Constableville, NY plus train ride to Canada!!!

(c)

Bad Driver of the Week 8/10/2012 - Car Almost Wrecks Honda CBR

(d)

Quinn on the bagpipes playing "Amazing Grace" with the Choir during Communion - SHS Graduation 2013

(e)

How to Dribble Faster | Basketball Moves

(f)

Figure 3: Examples of video and title pairs from Kinetics-210k.

descriptions. Figure 4c shows that the action in video is shot putting rather than *spinning* (appears in associated caption). Figure 4f illustrates that a person is climbing but its caption is mainly about high jumpping. Figure 4e shows that video content can also be noisy due to low video quality and shot transformation.

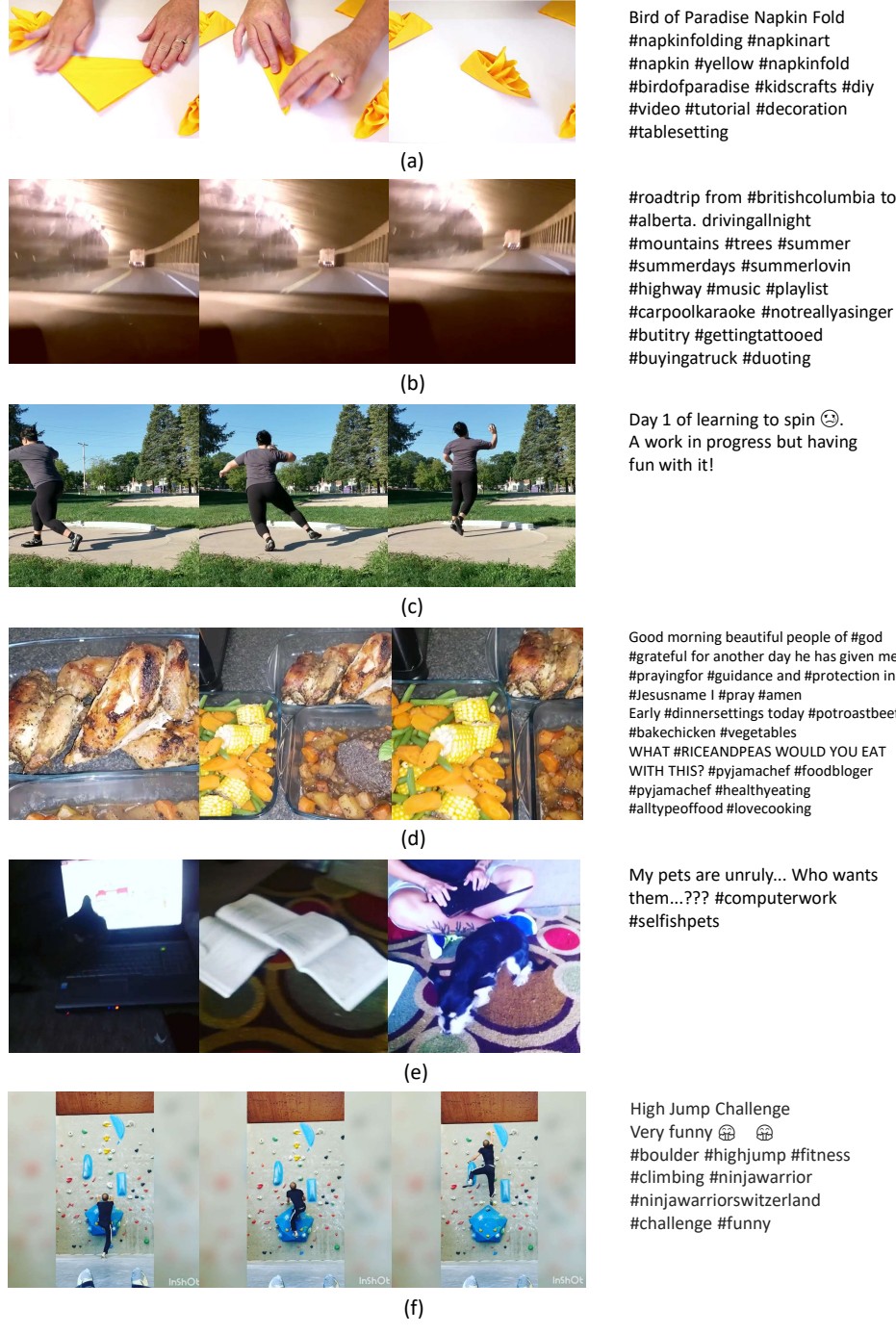

Figure 4: Examples of videos and their associated captions from Instagram-300k.

