# OpenReview forum: "Learning Spatiotemporal Features via Video and Text Pair Discrimination"
_ICLR.cc/2021/Conference — Reject_

### Official Review · AnonReviewer1 · 2020-10-28
**Good work showcasing video representations can be trained with much fewer data samples (on the order of 100K)**

**Rating:** 6
**Confidence:** 3

**Review:**

*Summary:*
The paper proposes an approach to learn a video feature backbone in an unsupervised manner through the use of video titles (text modality) associated with user generated content from Youtube or Instagram. The key idea is to use a contrastive loss that increases the similarity score between a positive pair vs. a negative pair. Contrary to previous works in this direction that require millions to hundreds of millions of paired clips, this work shows that good performance can be achieved by using much fewer (on the order of 100k) clips. The learned video model achieves good performance on standard action recognition datasets.

*Strengths:*
1. Shows that videos and their titles can be used to learn video features from scratch in an unsupervised manner.
2. Can achieve comparable performance with much less data in the scale of few 100K videos (rather than 100M clips as done in [Miech, et al. CVPR 2020]).
3. Evaluation on Kinetics-400, UCF-101, and HMDB shows comparable performance. Ablation studies are quite insightful (especially comparison between ranking loss vs. proposed CPD). It is also nice to see the performance difference by using kNN vs. classifier on Kinetics-400.

*Weaknesses:*
1. Novelty of the method itself (not the task to which it is applied) seems fairly limited. In fact, a majority of the model development resembles that of [Wu, et al. CVPR 2018]. However, to me, the motivation to arrive at cross-pair discrimination from this direction is unclear. Almost every paper on cross-modal learning in the last five+ years leverages some form of text feature, visual feature, and computes a dot-product between them (as indicated in the ranking loss). In such a case, what do we learn by approaching this problem using the self-instance discrimination which also performs very poorly? An additional complexity is with respect to the memory bank, but I believe this is only used for improving the estimation of the denominator in Eq. 3, right?

2. Experimental details:
(i) Some statistics about the datasets: Kinetics-210K and Instagram-300K would be nice to include. What is the average video duration? How many clips are obtained from each video? (the latter is especially important as HowTo100M also has about 1.x M videos, but 100+ M clips, albeit with separate text labels for each clip).
(ii) Why are YouTube videos considered PGC while Instagram UGC?
(iii) Since the temperature parameter is so specific, 0.07, I wonder how sensitive is the method to this parameter. An ablation with tau = [0.05, 0.1] would be nice, perhaps on the Kinetics-400?
(iv) Based on the answer to point (i), could the audio-based supervision from Audioset-1.8M be considered a comparable sized dataset?
(v) Since a major emphasis of the paper is on the smaller size of the data that allows learning similar performing representations, it would be nice to include additional details such as time taken to complete entire training on the 8 GPUs mentioned and perhaps an estimated comparison to larger datasets.

3. Minor points:
(i) Impact of curriculum learning is limited. While reading the article, it felt like this may play a large role - especially with sentences like "If we train both models simultaneously in the beginning, the random noise produced by video model will destroy the parameters of language model." Clearly, 0.9% difference (between freezing language model or not) in performance is not a destruction of language model parameters. Would request authors to please tone this down.
(ii) Typo in introduction: "So these expected these associated modalities ..." some text is repeated.
(iii) Sec 3.1, it should probably be i.e., f^v_i \in R^d and "f^t_i" \in R^d
(iv) Consider rephrasing grammar in the first sentence of the "Ranking loss." paragraph. Also, has "a" associated --> has "an" associated
(v) Typo: "that the these videos" below Instagram-300k.
(vi) Table 4.3 --> Table 2?
(vii) MIL-NCE (Stroud et al. 2020) --> MIL-NCE (Miech et al. 2020)

*Overall rating:*
I am generally in favor of accepting this paper given the comparable performance while requiring several orders of magnitude less dataset sizes, so that an academic lab may be able to train models. Among the open questions, I am curious about weakness point 1, and the inspiration to write this work from the point of view of self-instance discrimination.

---

> ### Author Response · Authors · 2020-11-23
> **Response to reviewer1**
>
> Q8. About motivation and inspiration from self-instance discrimination
>
> Cross-modal self-supervised learning is becoming popular for video learning during the past few years. Some methods directly propose a ranking loss as the training objective of cross-modal learning.  However, the ranking loss lacks a probability interpretation and fails to model uncertainty in a probabilistic manner.
>
> Our formulation framework generally follows the instance discrimination in image domain (Wu, et al. CVPR 2018) and extends it into the cross-modal scenario. However, the direct extension of instance discrimination with a joint distribution does not work as shown in Eq.(2).  Instead, in our proposed extension, we model the correlation between text and video with a conditional probability distribution as shown in Eq.(3), which can provide a more principled probabilistic explanation. As analyzed in Appendix A1, this probabilistic formulation can lead to a more reasonable loss function than the original ranking loss.
>
> The memory bank is not only used for estimating the denominator, but also for providing positive and negative samples when calculating Eq. (5).
>
> Q9. Experimental details
>
> (i) For Kinetics-210k, we directly use videos in Kinetics-400. So the duration of each video is 10 seconds. For Instagram-300k, the original videos are less than 60 seconds and we trim each video to 20 seconds from the middle of it for storage limitation. Since each video has one associated text information (title or caption), only one clip can be obtained from each video. (ii) Kinetics dataset is collected from Youtube and annotated carefully, so it is Profession Generated Content (PGC). Videos and captions of the Instagram dataset are uploaded by users and we collect this dataset without any manual filtering. Thus it is User Generated Content (UGC). (iii) The temperature of 0.07 is directly following [Wu, et al. CVPR 2018] and we didn't tune it. (iv) Base on point (i), the size of our datasets is still nearly one order of magnitude smaller than Audioset-1.8M (0.3M v.s. 1.8M).  (v) For Instagram-300k, it takes 10 days for completing the entire training on the 8 GPUs. For larger datasets, it might need one or two months. Also, the storage space can also be a large burden for larger datasets.

---

### Official Review · AnonReviewer3 · 2020-10-28
**Contributions are incremental**

**Rating:** 4
**Confidence:** 4

**Review:**

This paper concerns the problem of learning video representation from paired video-text pairs. The proposed framework is weakly-supervised as the text associated with videos comes from user-provided YouTube titles or Instagram captions. The proposed method uses standard visual encoder and textual encoder and similarity measurement for the joint embedding space. Overall, the paper is written in good clarity and has shown decent improvements over some of the existing methods. But the contributions are somewhat incremental considering the numerous existing/concurrent work built upon contrastive learning for video. The reasons are as follows.

i) Some of the claims are not well-substantiated. For example, on page one bottom, the paper claims that existing works learn "high-level visual-text embedding" as opposed to "video representation" as in this work, which is not true considering papers such as Miech et al., 2020.

Another notable claim in the paper is that "[...] Our work demonstrates that pre-training a relatively small video-text dataset is also possible to match the STOA performance [...]", which is somewhat over-stated given that the pre-training data is curated on a quite restricted domain (e.g., human actions) and sometimes containing manually-curated data, and could intuitively benefit downstream tasks about the same domain (all tested datasets fall into this category) vs. instructional or more generic pre-training data. Besides, there are essential questions that remain to be answered, such as whether bringing in more data could further benefit the model.

ii) The technical contributions and empirical results are incremental (e.g., Tab. 3). Some experimental results are not comprehensive, for instance, Tab. 2 should involve SOTA methods evaluated under the same setting to demonstrate model effectiveness.

Other minor comments:

i) PGC and UGC on page two are mentioned before defining.

ii) It's unclear what f^v and f^t are in Eq. 1. How are they related to f_i^v and f_i^t.

iii) What does the Tab. 4.3 on page seven refers to?

---

> ### Author Response · Authors · 2020-11-23
> **Response to reviewer 3**
>
> Q6. About our statement
>
> Thanks for your comment.  We want to remind you that our pre-trained dataset Instagram is uncurated and results on Instagram are very comparable to that of pre-training on Kinetics.
>
> About the data domain, we agree that currently these videos might be more relevant to human actions and in the future we might broaden the domain of our pre-trained datasets.  In fact, every dataset has its own data bias and sometimes these data bias is useful for specific domain application. It is really difficult to generate a universal dataset, that works well for any down-stream task.
>
> Finally, we want to emphasize that the data distribution of our Instagram is very different from that of UCF101, HMDB51, and Kinetics. The videos in our pre-trained dataset (Instagram) is User-Generated Content with large variations and diversity. Transferring from UGC videos to PGC videos (moive, TV etc.) is more challenging.
>
>
>
> Q7. More results on Tab. 2
>
> Thanks for your comment. We agree that self-supervised methods should not merely report the results of transferring to small datasets, e.g. UCF101 and HMDB51, since hyper-parameters adopted during fine-tuning have a great influence on the results. Thus, we propose to also report the classification accuracy of k-Nearest Neighbor (kNN) and linear classifier (LC) on the frozen feature. This is more straightforward to verify the effectiveness of learned representation.
>
> We would like to evaluate other methods under this setting. We download the pre-trained S3D model of MIL-NCE and then extract 1024 dimensional features after the average pooling. Then, we conduct kNN and LC and obtain the top-1 accuracy of 49.5 and 60.1 respectively. Our CPD methods show competitive performance to these results, but we pre-trained on a relatively small dataset. We will update this result in the revision.

---

### Official Review · AnonReviewer4 · 2020-10-29
**This paper proposes a method for weakly supervised video representation learning. In specific, this paper utilizes the paired relationship between web video and its associated text caption, a cross-modal pair discrimination framework is proposed to encourage high similarity between positive video-text pairs while low-similarity between negative pairs.**

**Rating:** 5
**Confidence:** 5

**Review:**

Advantage of this paper:
1.	This paper is well motivated and well written and the topic of this paper is valuable.
2.	It’s interesting to utilize text as weak supervision for video representation learning, and the experiment results also indicate effectiveness of the learned video representation.

Weakness of this paper
1.	The novelty of this paper might be limited. Previous works have explored the possibility of utilizing text as weak supervision for video representation learning (MIL-NCE), from the reviewer’s perspective, the main difference is that the different loss function is adopted.
2.	Compared with methods that adopt other information (such as audio) as weak supervision, there is an inherent advantage of using text as supervision since pretrained text models such as BERT can be utilize as a guidance. So a meaningful comparison would be the comparison with TWS and MIL-NCE, although the proposed method can achieve comparable performance with other methods with much less data, the author does not give analysis about what design in the proposed method that enables this.
3.	The performance comparison is not convincing enough. From Table 3, we can see that different backbones are used for different methods, the reviewer worries that the superiority of the proposed method might be brought by a stronger backbone.

---

> ### Author Response · Authors · 2020-11-23
> **Response to reviewer 4**
>
> Q3.  About novelty with MIL-NCE.
>
> First it should be noted that our work is concurrent with MIL-NCE as stated in related work (In fact, the preliminary version of our work is submitted to CVPR 2020 as well and we release the CVPR version on arXiv from then. )
>
> Second, although the basic idea of our work is very similar to MIL-NCE, there is an important technical difference between ours and MIL-NCE. For MIL-NCE, they simply search for negative pairs in each batch, whose negative candidates are greatly limited by the batch size, while our CPD builds a memory bank to store the negative pairs over the entire dataset. For self-supervised learning, the negative pairs play important roles in final representation learning.
>
> Finally, our CPD demonstrates that pre-training a relatively small dataset is able to yield a comparable performance to those methods of using order magnitude more data, which is meaningful and practicable for the scenarios with limited computational facilities. We think, for a university lab, it is almost impossible to reproduce the results of MIL-NCE by training from 100M clips with a single workstation of 8 GPUs.
>
>
>
> Q4. What design enable effectiveness.
>
> The ablation studies show the effectiveness of our method in four technical aspects: memory bank for a large number of negative pairs, loss function, training strategies and textual encoders.  A large number of negative pairs and the cross-pair discrimination loss function gives a better performance than ranking loss which is used in TWS. Also, the curriculum learning strategy is helpful to deal with the random noise produce by the video model, which is not used in both TWS and MIL-NCE.
>
> Q5. Different backbones.
>
> Thanks for your suggestion. We would like to test our method using other backbones. We pre-train our model using S3D as the backbone network on Instagram-300k dataset. It shows a similar performance with 3D ResNet50 (89.4% when transferring to UCF101). We will update this result in the revision.

---

### Official Review · AnonReviewer2 · 2020-10-29
**An OK paper: marginal novel, unfair comparisons and missing baselines**

**Rating:** 4
**Confidence:** 5

**Review:**

The paper proposes a weakly supervised method for learning spatiotemporal features by video and text pair discrimination, namely cross-modal pair discrimination (CPD). This can be considered as an extension of (Wu et al. 2018) to video and text. On technical perspective, the original method Wu et al. is applied on images, while CPD is applied on video and text (video's title for Kinetics or hashtag search for Instagram). The most novel technical contribution of this paper is making Wu et al. 2018 cross-modal (between video and text). However, compared with Wu et al. 2018, it requires more supervision (weakly supervised vs. unsupervised). On the experiments, some comparisons are unfair and some experimental setups are biased (detail below).

(+) Pros
- Making Non-Parametric Instance Discrimination (Wu et al. 2018) cross-modal is interesting.

(-) Cons
- Novelty is marginal. As mentioned above the main contribution is making Non-Parametric Instance Discrimination cross-modal. But at the same time, the weakly supervision setup make it less significant compared to the original work. Moreover, cross-modal has been used before for cross-model between audio-visual (L3Net, AVTS, XDC), video-speech (Sun et al. 2019b, Meich et al. 2019). (The reviewer is open to this level of novelty if the experiments are solid, see next for comments on experiments).

- Experiments contains some unfair comparisons and some experimental setups are biased.
1. Table 3 compares CPD with other self-supervised methods. Note that CPD is a weakly supervised method. The right one to compare with CPD may be Ghadiyaram et al. CVPR'19. The author(s) may argue that this work used a lot more data than CPD. That is true, but they can train a baseline network, e.g. same backbone as CPD, with a cross-entropy loss on Kinetics-title-clean and Instagram-300k using title or search query as label (similar to Ghadiyaram et al. CVPR'19). This is a fair and important baseline to understand the proposed method of CPD.
2. There are potentially biased setup in Kinetics experiments. Recall how Kinetics was collected and annotated [Kay at et. 2017]. Videos are searched and retrieved from Youtube using Kinetics-taxonomies (mostly contain a verb followed by a noun), then further verified by human annotator. This means if this paper uses titles of the Kinetics videos, it is very likely that it has the correct verb+noun combination of the ground-truth taxonomy for that video. Moreover, not sure if the authors use the temporal annotations (10 second segment of video) when they sample clips from Kinetics videos (the paper mention about how long the clips are and temporal striding, but never mention if they ignore or use Kinetics temporal annotation).

*Some minor comments:
- abstract: pre-training a relatively small dataset -> pre-training on a relatively small dataset
- introduction:  So these expected these associated -> So these associated
- missing i, j indices for f^u, f^t in Eq(1)? Similarly for Eq(2)?









Wu et al. Unsupervised Feature Learning via Non-Parametric Instance Discrimination, CVPR 2018.

---

> ### Author Response · Authors · 2020-11-23
> **Response to reviewer 2**
>
> Q1. More discussion about our method and comparison to Ghadiyaram et al. CVPR'19.
>
> We disagree with your opinion that our method should be compared with those weakly supervised methods rather than those self-supervised methods.
>
> First, we argue that our basic motivation and technical solution is totally different from the weakly-supervised methods (Ghadiyaram et al. CVPR'19). Although our method uses web text for supervision, there is an essential difference between ours and Ghadiyaram et al. CVPR'19. They use the hashtags as labels, which are generated by the search engine (not naturally with video data). These search engines could be built based on rich vision models trained from plenty of human supervision. In fact, the work of Ghadiyaram et al. CVPR'19 is performing knowledge distillation from a super search engine model, instead of training from the information of the data itself.
>
>
>
> On the other hand, our method aims to learn from the multi-modal information of data itself. These text data naturally concurs with these video data. In this sense, it is uploaded on the web with the video at the same time (such title or comments). This text information should be viewed as an inherent modality with web video, which is independent of the search engine model. From this perspective, our method aims to learn video representation from the data distribution itself and is similar to those cross-modal self-supervised learning methods.
>
>
>
> We really hope the readers could understand the subtle yet fundamental difference between our method from those learning from web query as supervision (one is knowledge distillation from a search engine model, while one is self-supervised learning from multi-modal information of data itself).
>
>
>
> Q2. About Kinetics dataset.
>
> We pre-train our model on the Kinetics-210k dataset because of the common practice of previous self-supervised video representation learning and cross-modal video representation learning methods ( (Korbar et al., 2018, Alwassel et al., 2019, Han et al., 2019, Sun et al., 2019a).  We agree that the Kinetics-210k is a biased and curated dataset. Thus, we further conduct experiments on uncurated Instagram-300k dataset.  The model pre-trained on Instagram-300k shows competitive performance to Kinetice-210k when transferring to UCF101 and HMDB51, which indicates that our CPD model can handle real-world noisy videos.
>
>
>
> In fact, in addition to the ablation study, we all report the performance of training from the Instagram dataset in other tables.

---

### Decision · Program_Chairs · 2021-01-07
**Final Decision**

**Decision:**

Reject

**Comment:**

The paper presents an approach for weakly supervised pre-training for videos using textual information provided with web videos on Youtube and Instagram.

## Strength
* The work shows strong results with relative small dataset and computational resources compared to other work in the area of self/weakly supervised learning for videos.
* Interesting ablations

## Main Concerns
* The authors don't discuss and compare to the weakly supervised work [Ghadiyaram et al. CVPR'19] adequately. Furthermore, the authors characterize the work incorrectly in their author response as detailed by R2. I agree to R2 here and like to highlight the concern is not that the method of [Ghadiyaram et al. CVPR'19] being similar to this work but the level/type of supervision.
* Limited novelty over prior work.

## Further Concerns
* Some unclarities
* The authors did not provide an updated revision of the pdf

Overall the paper received reject and borderline scores after author response and discussion (With the strongest score 6 from R1) due to the concerns concerns listed above apart from the ability to work with small number of data. I think the missing comparison to  [Ghadiyaram et al. CVPR'19] which operates in a similar setting weights strongly and I recommend reject.